

# Tropical cyclones vertical structure from GNSS radio occultation: an archive covering the period 2001–2018

**Elżbieta Lasota[1,2], Andrea K. Steiner[3,4], Gottfried Kirchengast[3,4], and Riccardo Biondi[2]**

[1]Institute of Geodesy and Geoinformatics,
Wrocław University of Environmental and Life Sciences, Wrocław, 50356, Poland
[2]Dipartimento di Geoscienze, Università degli Studi di Padova, Padua, 35131, Italy
[3]Wegener Center for Climate and Global Change (WEGC), University of Graz, Graz, 8010, Austria
[4]Institute for Geophysics, Astrophysics, and Meteorology/Institute of Physics, University of Graz,
Graz, 8010, Austria

**Correspondence:** Riccardo Biondi (riccardo@biondiriccardo.it)

**Abstract.** Tropical cyclones (TC) are natural destructive phenomena, which affect wide tropical and subtropical areas every year. Although the correct prediction of their tracks and intensity has improved over recent years, the knowledge about their structure and development is still insufficient. The Global Navigation Satellite System (GNSS) radio occultation (RO) technique can provide a better understanding of the TC because it enables us to probe the atmospheric vertical structure with high accuracy, high vertical resolution and global coverage in any weather conditions. In this work, we create an archive of co-located TC best tracks and RO profiles covering the period 2001–2018 and providing a complete view of the storms since the pre-cyclone status to the cyclone disappearance. We collected 1822 TC best tracks from the International Best Track Archive for Climate Stewardship and co-located them with 48 313 RO profiles from seven satellite missions processed by the Wegener Center for Climate and Global Change. We provide information about location and intensity of the TC, RO vertical profiles co-located within 3 h and 500 km from the TC eye centre, and exact information about temporal and spatial distance between the TC centre and the RO mean tangent point. A statistical analysis shows how the archive covers all the ocean basins and all the intensity categories well. We finally demonstrate the application of this dataset to investigate the vertical structure for one TC example case. All the data files, separately for each TC, are publicly available in NetCDF format at https://doi.org/10.25364/WEGC/TC-RO1.0:2020.1 (Lasota et al., 2020).

## 1 Introduction

The tropical cyclones (TCs), known also as hurricanes in the North Atlantic Ocean and northeast Pacific, typhoons in the northwest Pacific, and simply as cyclones in the South Pacific and Indian Ocean, are extreme weather events affecting the social lives of many people and the economy of entire countries. The understanding of the development of TCs have increased with the availability of satellite measurements, but a decisive improvement was given in the last decade by the use of the Global Navigation Satellite System (GNSS) radio occultation (RO) technique allowing us to profile the atmo-sphere with high vertical resolution and high accuracy (Anthes et al., 2008). The GNSS RO technique uses GNSS signals and low Earth orbit (LEO) receivers to profile the atmospheric refractivity, from which profiles of temperature, pressure and humidity (Kursinski et al., 1997) are retrieved in the moist atmosphere by using background information. The RO technique was developed for observing the Earth's atmosphere and climate (Anthes et al., 2008; Steiner et al., 2011). It became important for the analyses and forecast of extreme atmospheric events (Bonafoni et al., 2019) motivat-

ing the interest and the launch of several new public and private missions (Cirac-Claveras, 2019).

Cardinali (2009) showed the high impact of RO to improve weather forecast, especially in remote areas of the globe where no other instruments are available with high vertical resolution. Several studies demonstrated the impact of RO profiles to improve the TC track forecast with assimilation in numerical weather prediction (NWP) models. Huang et al. (2005), for the first time, assimilated RO refractivity profiles to forecast the typhoons Nari in 2001 and Nakri in 2002, which developed in the northwestern Pacific Ocean. This study was followed by many others, focusing on cyclone events: Hurricane Ernesto in 2006 (Liu et al., 2012), Typhoon Usagi in 2007 (Kunii et al., 2012), typhoons Jangmi in 2008, Hagupit in 2008 and Sinlaku in 2008 (Hsiao et al., 2012), Super-cyclone Gonu in 2007 (Anisetty et al., 2014), and Tropical Cyclone Phailin in 2013 (Hima Bindu et al., 2016). Huang et al. (2010) were the first who analysed a complete TC season followed by Chen et al. (2015). In a recent study, Chen et al. (2020) assimilated RO data during the genesis of 10 TCs in the northwestern Pacific Ocean in the period 2008–2010. The results confirmed the benefit of assimilation of RO refractivity improving the humidity estimation in the lower and the middle troposphere and thus the forecast of the TCs. However, RO data are now widely used to study the TCs' structure and their impact on the surrounding atmosphere. Biondi et al. (2011a) investigated for the first time the vertical structure of Typhoon Hondo in 2008 and Hurricane Bertha in 2008 and found a clear signature of the TC cloud top height. Biondi et al. (2011b) demonstrated that the presence of a TC creates a large positive bending angle anomaly in the upper troposphere and lower stratosphere (UTLS) corresponding to the TC anvil cloud top. The validation of these results with radiosondes and the Cloud-Aerosol Lidar with Orthogonal Polarization (CALIOP) revealed that the bending angle can be used to detect the cloud top (Biondi et al., 2013) and the possible overshooting (Biondi et al., 2015). In a more recent study, Lasota et al. (2018) analysed the RO bending angle sensitivity to the presence of clouds in TCs showing a significant signature of clouds between 8 and 14 km of altitude.

With the advent of the RO, the scientific community was able to better understand the TC inner thermal structure and water vapour (WV) content at different layers. The TC track and intensity predictions, until 2006, were almost completely based on parameters such as surface temperature, cloud top temperature and surface winds at the outer radius (Brueske and Velden, 2003; Demuth et al., 2004; Dvorak, 1975; Kidder et al., 1978; Velden et al., 2006) from different remote sensing techniques such as infrared and microwave sounders and imagers (King et al., 1992), lidar (Poole et al., 2003), reflected light polarization (Knibbe et al., 2000), and oxygen A-band technique (Koelemeijer et al., 2002). Anthes et al. (2003) were the first, comparing RO soundings with radiosonde observations during the Typhoon Toraji

in 2001. The results revealed that the RO temperature profiles were consistent within 1 K, whilst RO water vapour observations tended to be slightly drier than radiosonde measurements above the middle troposphere. A similar agreement between RO and dropsonde observations was presented by Anthes (2011) for the Typhoon Jangmi in 2008. Anthes et al. (2008) demonstrated the importance of RO temperature and WV assimilation to forecast Hurricane Ernesto in 2006 (Chen et al., 2014; Liu et al., 2012). Winterbottom and Xiao (2010) showed that the quality and the horizontal resolution of RO was high enough to study TCs, even before the Constellation Observing System for Meteorology, Ionosphere and Climate (COSMIC) six-satellite mission was launched. However, thanks to the higher number of RO observations provided from COSMIC after 2006, it has been possible to get a better understanding of the TCs' thermal structure: a warm core in the troposphere (Zou and Tian, 2018), a cooling corresponding to the TC anvil top height (Biondi et al., 2013; Rivoire et al., 2016) and an increase in WV in the lower stratosphere (LS) above the outermost rainbands (Venkat Ratnam et al., 2016). Vergados et al. (2013) for the first time used more than 1500 RO temperature, water vapour and refractivity profiles to study the moist thermodynamic structure in the lower and the upper troposphere of 42 North Atlantic TCs in the period 2002–2010. The analysis showed that the RO observations are able to capture the dimension, eyewall and rainbands of the TCs at different stages. In particular, the gradual decrease and wavelike pattern of water vapour was observed with increasing distance from the TC centre. Furthermore, a drop of WV was noticed in the lower and upper troposphere when the TCs develop from a tropical depression to Category 1 intensity.

The tropopause layer is often affected by the presence of the TC, and this is easily detectable by using the RO profiles. In particular, the high vertical resolution of RO profiles shows that the TC anvil top generates a double tropopause effect when it does not reach the tropopause level (Biondi et al., 2011b, 2013; Vergados et al., 2014) and the tropical tropopause layer (TTL) thickness is reduced (Ravindra Babu et al., 2015; Venkat Ratnam et al., 2016). Deep convective towers, usually developed within the TC eyewall and rainbands, generate gravity waves (GWs) transporting energy to the upper atmosphere. The high vertical resolution of RO can reveal the GW spectral characteristics (Chane Ming et al., 2014) associated with the presence of the TC and show how the intensification of the TC creates LS GWs (Chane Ming et al., 2014; Rakshit et al., 2018). A comprehensive review on the use of RO observations to study TCs is given by Bonafoni et al. (2019).

GNSS RO data are processed by several processing centres (Danish Meteorological Institute – DMI, EUMETSAT, German Research Centre for Geosciences – GFZ, Jet Propulsion Laboratory – JPL, University Corporation for Atmospheric Research – UCAR, Wegener Center for Climate and Global Change – WEGC) each using a different processing scheme.

Regular inter-comparison studies of RO products from different centres (Ho et al., 2009; Steiner et al., 2013) are performed to improve the data and to understand differences. Latest results showed that RO data from different processing centres are highly consistent in the UTLS, and differences become larger above 25 km altitude (Steiner et al., 2020). In this work, we use the WEGC RO dataset.

The aim of this work is to provide a comprehensive archive covering the period 2001–2018 collecting all the available information about TCs together with co-located RO observations to be used as a background for future studies to improve the knowledge of TC structure and development, to better understand the pre-TC environment, and to study the effect of TCs in the UTLS structure. For each TC, the information about track and intensity is combined with all the RO vertical profiles available within 500 km and 3 h. The paper describes the datasets used to create the archive, explains the methodology to co-locate the different datasets, shows the statistical analysis of data spatial distribution, highlights an example of possible use of the dataset, and finally remarks the uncertainties and capabilities.

## 2 Data and methods

### 2.1 RO profiles

We have used the GNSS RO products level 1b (L1b) and level 2 (L2) processed by the Wegener Center for Climate and Global Change (WEGC) through the Occultation Processing System (OPS) version 5.6, which use University Corporation for Atmospheric Research (UCAR) version orbit and phase data (Schwärz et al., 2016; Angerer et al., 2017, see Table 1). Out of this archive, we have selected the data of the CHAllenging Minisatellite Payload (CHAMP) from 2001 to 2008 (Wickert et al., 2001); the Satélite de Aplicaciones Científicas (SAC-C) form 2001 to 2013 (Hajj et al., 2004); the Gravity Recovery and Climate Experiment A (GRACE-A) from 2007 to 2017 and GRACE-B from 2014 to 2017 (Beyerle et al., 2005); the Constellation Observing System for Meteorology, Ionosphere and Climate (COSMIC) from 2006 to 2018 (Anthes et al., 2008); the Meteorological Operational satellite (MetOp) from 2008 to 2018 (Luntama et al., 2008); and the Communication/Navigation Outage Forecasting System (C/NOFS) from 2010 to 2011 (de La Beaujardière, 2004). The WEGC RO OPS v5.6 product includes vertical profiles of various variables including specific humidity, temperature, refractivity and bending angle of the atmosphere with 100 m vertical sampling from near the surface altitude up to 60 km height with global coverage.

In the regions where the water vapour is negligible (usually above 10 km of altitude), the refractivity profiles can be transformed in dry temperature and dry pressure profiles by using a reduced refractivity equation (Scherlling-Pirscher et al., 2011). In the lower troposphere, the abundant amount of water vapour makes the dry air assumption not valid, and

ancillary information from the weather model is required to retrieve the physical atmospheric parameters. The details of the OPS v5.6 tropospheric retrieval scheme is described by Li et al. (2019) introducing the moist air retrieval algorithm, inter-comparing it with the UCAR/COSMIC and EU-METSAT Radio Occultation Meteorology Satellite Application Facility (ROM SAF) retrievals, and showing that in the lower to middle troposphere the moisture information is predominantly coming from the RO data. These results are also confirmed by Rieckh et al. (2018), who inter-compared tropospheric humidity profiles from four retrievals (including OPS v5.6) with radiosondes.

Furthermore, we made use of global monthly mean multi-satellite climatologies processed by the WEGC (based on OPS v5.6 profiles in the period 2001–2017). The climatological profiles of bending angle, specific humidity and temperature are available with $2.5° \times 2.5°$ horizontal resolution.

### 2.2 TC tracks

In this work, we focused on the TCs that occurred in the period 2001–2018 overlapping with the RO data availability. The comprehensive information of TC best track data was downloaded from the International Best Track Archive for Climate Stewardship (IBTrACS) version 04 (Knapp et al., 2010, 2018). The IBTrACS collects and combines the best track data from each World Meteorological Organization (WMO) Regional Specialized Meteorological Center (RSMC) and Tropical Cyclone Warning Center (TCWC), but also from other meteorological agencies, who trace the TCs in the regions of the interest.

In this dataset, we store the wind speeds and central pressures obtained from the WMO responsibility agency for the particular ocean basin. The RSMC and the TCWC participate in the Tropical Cyclone Programme (World Meteorological Organization, 1980) and are officially required to forecast and report the information about TC position, movement and intensity in the designated area of responsibility (Table 1).

The IBTrACS dataset is disseminated in different formats, CSV, netCDF or shapefile formats, for various subsets such as for separate ocean basins, time periods, or for all TCs in the record. The archive is based on post-seasonal reanalyses and comprises information about storm name, position, maximum sustained wind speed or minimum central pressure, mostly reported with 6 h temporal resolution, as well as some additional parameters interpolated to 3 h resolution. The RSMC and TCWC compute and average the maximum sustained wind speed in different periods and hence cannot be directly compared. The US agencies use a 1 min averaging period, the Indian RSMC uses a 3 min averaging period, whilst the rest of RSMC and TCWC (Brisbane, La Réunion, Nadi, Tokyo, Wellington) use a common 10 min averaging period. The IBTrACS does not perform any wind speed transformations and provides original data from each agency, leaving to the users the choice of method for inter-

**Table 1.** The list of main agencies included in the IBTrACS dataset in the different ocean basins: North Atlantic (NA), eastern North Pacific (EN), western North Pacific (WP), north Indian (NI), south Indian (SI), South Pacific (SP), South Atlantic (SA).

| Agency | Abbreviation | Ocean basin |
|---|---|---|
| National Hurricane Center (NHC) of National Oceanic and Atmospheric Administration | hurdat_atl | NA |
| (NOAA, USA) as RSMC Miami | hurdat_epa | EN |
| Japan Meteorological Agency as RSMC Tokyo | tokyo | WP |
| India Meteorological Department as RSMC New Delhi | newdelhi | NI |
| Météo-France as RSMC La Réunion | reunion | SI |
| Australian Bureau of Meteorology as TCWC Perth, Darwin, Brisbane | bom | SI, SP |
| Meteorological Service of New Zealand, Ltd as TCWC Wellington | wellington | SP |
| Fiji Meteorological Service as RSMC Nadi | nadi | SP |
| Automated Tropical Cyclone Forecasting System for U.S. Department | atcf | SA, NA, EP |
| of Defense and National Weather Service TCWC | | |

agency comparison. However, for the statistics and analyses presented in this paper, we follow the guidelines given by the WMO (Harper et al., 2010). Conversion factors between the 10 and 3 min sustained wind speed into 1 min wind speed are calculated using the equation E-2 from the World Meteorological Organization (WMO) "Guideline for converting between various wind averaging periods in tropical cyclone conditions" (Harper et al., 2010). Next, to unify reported wind speeds to 1 min sustained wind speed, the original wind speeds are multiplied by the calculated conversion factors of 1.08 and 1.05 for 10 and 3 min averaging periods, respectively. The resulting values are used as a reference to categorize the TC intensity according to the commonly used Saffir–Simpson hurricane scale (Simpson, 1974), which identifies seven levels based on the wind speed (Table 2).

## 2.3   Co-location of TC and RO observations

Retrieving RO profiles demands the appropriate knowledge of geometry between the low Earth orbit (LEO) receiver and GNSS satellites, which results in the random distribution of profiles in the time and space. Furthermore, the retrieved tangent point trajectory is curved and diverges from the vertical line since the GNSS and LEO satellites move with different speeds on non-coplanar orbits (Foelsche et al., 2011). In this work, we use the latitudes and longitudes of mean tangent points provided in the WEGC RO products. We co-locate each TC best track position with RO profiles which occurred within 3 h and 500 km from the TC eye centre. The temporal window has been chosen as half temporal resolution of the TC best track reports, while the space window is chosen as a commonly used average TC radius of influence (Barlow, 2011; Knaff et al., 2013), which also corresponds to the half maximum path covered by a TC in 6 h. In fact, we have computed in our dataset maximum, minimum and average distances covered in 6 h by a TC as 969.2, 0 and 110.4 km, respectively. Thus, a single RO profile could be co-located with more than one TC best track position. In this case, we

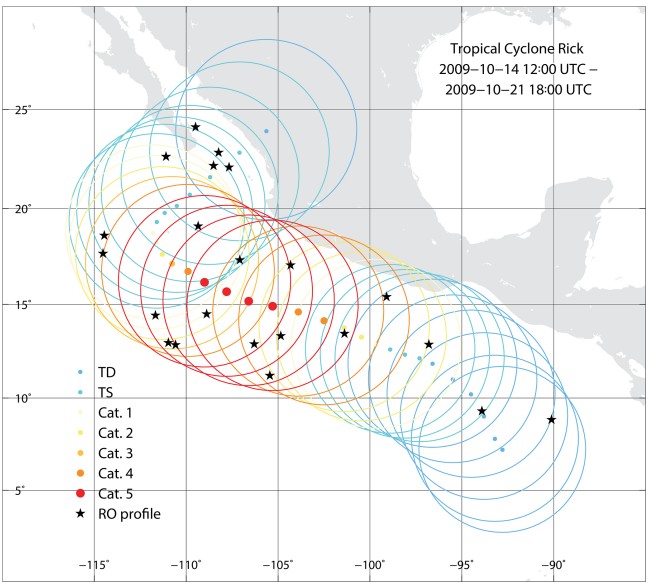

**Figure 1.** An example of co-location of TC with RO profiles (black stars) based on Hurricane Rick developed between 14 and 21 October 2009. Dots present the TC eye position, whilst circles mark the 500 km co-location criterion. Colours indicate the intensity of TC.

classified it to each TC track position, which meets the spatial and time condition.

As an example, we report in Fig. 1 the best track of Hurricane Rick in 2009, which developed from 14 October 2009 to 21 October 2009 in the eastern Pacific Ocean basin close to the Mexican shore. The dots represent the TC eye centre, the circle indicates the 500 km radius that we have chosen as reference and the colours show the TC intensity (from TD in blue to Category 5 in red). The stars denote the position of the co-located RO mean tangent point, and it becomes clear how a RO profile can be associated with more than one TC stage.

https://doi.org/10.5194/essd-12-1-2020

**Table 2.** TC intensity based on the Saffir–Simpson Hurricane Wind Scale.

| Category | Tropical depression (TD) | Tropical storm (TS) | Category 1 (Cat. 1) | Category 2 (Cat. 2) | Category 3 (Cat. 3) | Category 4 (Cat. 4) | Category 5 (Cat. 5) |
|---|---|---|---|---|---|---|---|
| 1 min maximum sustained wind speed ($\mathrm{m\,s^{-1}}$) | $\leq 17$ | 18–32 | 33–42 | 43–49 | 50–58 | 58–70 | > 70 |

## 2.4 Data structure of the archive

For each TC, at every single step reported by the IBTrACS, we store the information about the co-locations between the TC and the RO, the vertical structure of the TC provided by RO data, and the background environment (Table 3). The TC is described by the basin of development; the name of the responsible recording WMO agency; the distance of the TC from land; the date, time and coordinates at each 6 h best track stage; the nature of the storm; the storm translation speed; the minimum central pressure; and the maximum sustained wind speed provided by the responsible WMO agency and stored in wmo_pres and wmo_wind variables, respectively. The co-locations between TCs and ROs are detailed with date, time and coordinates of the RO; the temporal difference between the co-located RO profile and the TC best track time; and the spatial distance between the RO mean tangent point and the TC best track coordinates. The TC vertical structure is given by the vertical profile from the surface (0 km) to 60 km with 10 m sampling for specific humidity, pressure, temperature, refractivity and bending angle. As a reference, we also report the climatological profiles of specific humidity, temperature and bending angle in the same area in order to compute the vertical anomaly structure with respect to the climatology.

The NetCDF format has been developed to share the array-oriented scientific data; therefore, the structure of the TC-RO archive is arranged to fulfil the array structure requirement. All the data are stored in the up to 3D arrays with particular dimensions: $N_{\mathrm{TC}}$, $N_{\mathrm{maxRO}}$ and $N_{\mathrm{alt}}$. $N_{\mathrm{TC}}$ refers to the total number of the TC best track positions, separately for each analysed TC, whilst $N_{\mathrm{maxRO}}$ corresponds to the maximum number of co-located RO profiles with a single TC track position. Since not every TC track position has as many RO co-locations as the value of $N_{\mathrm{maxRO}}$, the variables such as latRO, lonRO, bending angle and others may contain empty values, which are filled with appropriate filling values. The dimension, $N_{\mathrm{alt}}$, is the number of available RO vertical levels (refractivity, pressure, temperature or specific humidity) and by default is equal to 600.

All the details of data structure are reported in the Supplement.

## 3 Results

We have collected 48 313 co-locations between ROs and TCs from 1570 TCs, with at least one profile for 86 % of the TCs

occurring in the period 2001–2018 (1822 in total, Table 4). In the early period 2001–2006, the number of co-locations is limited to a few hundred because only CHAMP and SAC-C were in orbit. CHAMP started measuring in May 2001, and thus the year with the lowest number of co-locations was 2001 (only 50). In 2006, the COSMIC six-satellite constellation was launched, and the number of co-locations increased to some thousands per year. The year with the largest number of co-locations was 2008, with 5482 coming from 99 TC tracks. The highest number of co-located profiles comes from the MetOp-A receiver (Table 5) due to the largest time range availability (11 years). The ocean basin with the highest number of co-locations (Table 6) is the western Pacific, due to the larger number of TCs which are lasting for a longer period than the other ocean basins.

The co-locations are well distributed in all the ocean basins (Fig. 2), with a small number very close to the TC eye centre (172 in total), 1793 co-locations very close to or into the eyewall, and an increasing number moving away from the centre (Table 6). Since the TCs have different intensity and different characteristics according to the area where they develop (Biondi et al., 2015), we also report the statistics by ocean basin and by categories (Table 6 and Fig. 3). The western Pacific Ocean basin has the largest number (14 310) of co-locations, covering all the categories except Cat. 5 well. In the eastern Pacific Ocean and North Atlantic Ocean basins, we found the largest number of co-locations for the highest intensity Cat. 5 (14).

Demonstrating the use of the provided archive for an example case study, we report in Fig. 4 the case of Typhoon Hondo in 2008, which developed in the south Indian Ocean and reached the maximum intensity of Cat. 4. Hondo started as a tropical depression on 2 February 2008. Two days later it intensified to a TS and quickly reached TC intensity (reddish dots Fig. 4a) on 5 February 2008. The status of a TC persisted for 5 d, and then it weakened to a TD until the end of its life on 29 February 2008. Hondo is the TC for which we found the highest number of RO co-locations (Fig. 4a, black stars), with a total of 212 profiles, just 2 into the TC eye, 10 close to the eyewall, and 200 distributed between 100 and 500 km from the TC centre. Thirty-eight profiles are co-located with the TC status, 15 with TS and 159 with TD. The maximum number of profiles for a single stage was four, co-located with the TD on 12 February 2008. Figure 4b shows the temperature profile evolution with the time and altitude. The black profiles mostly indicate the TC stages and

**Table 3.** Parameters stored in the dataset files for each TC separately. $N_{\mathrm{TC}}$ denotes the number of TC track positions, $N_{\mathrm{alt}}$ denotes the number of altitude levels (600 by default) and $N_{\mathrm{maxRO}}$ stands for the maximum number of RO profiles found for a single TC best track position.

| Parameter (unit) | Dimension | Description |
| --- | --- | --- |
| altitude (m) | $N_{\mathrm{alt}} \times 1$ | Altitudes above geoid between 0 and 59.9 km with 100 m spacing. |
| latTC (°N) | $N_{\mathrm{TC}} \times 1$ | Latitude of current TC track position. |
| lonTC (°E) | $N_{\mathrm{TC}} \times 1$ | Longitude of current TC track position. |
| basin | $N_{\mathrm{TC}} \times 1$ | Flag values (1–7) indicating the ocean basin for the current storm position: 1 = east Pacific, 2 = North Atlantic, 3 = north Indian, 4 = South Atlantic, 5 = south Indian, 6 = South Pacific, 7 = western Pacific. |
| dist2land (km) | $N_{\mathrm{TC}} \times 1$ | Distance between current TC position and land. |
| landfall (km) | $N_{\mathrm{TC}} \times 1$ | Minimum distance of TC to land over next 3 h (0 means landfall). |
| nature | $N_{\mathrm{TC}} \times 1$ | Flag values (1–6) indicating the nature of the current TC stage: 1 = not reported, 2 = disturbance, 3 = tropical system, 4 = extratropical system, 5 = subtropical system, 6 = mixed (occurs when agencies reported inconsistent types not reported). |
| storm_dir (°) | $N_{\mathrm{TC}} \times 1$ | Storm translation direction. |
| storm_speed (m s$^{-1}$) | $N_{\mathrm{TC}} \times 1$ | Storm translation speed. |
| subbasin | $N_{\mathrm{TC}} \times 1$ | Flag values (1–9) indicating ocean sub-basin for the current storm position: 1 = Arabian Sea., 2 = Bay of Bengal, 3 = central Pacific, 4 = Caribbean Sea, 5 = Gulf of Mexico, 6 = North Atlantic, 7 = eastern Australia, 8 = western Australia, 9 = no sub-basin for this position. |
| wmo_agency | $N_{\mathrm{TC}} \times 1$ | Flag values (1–10) indicating name of the responsible WMO agency: 1 = not provided, 2 = atcf, 3 = bom, 4 = hurdat_atl, 5 = hurdat_epa, 6 = nadi, 7 = newdelhi, 8 = reunion, 9 = Tokyo, 10 = wellington. |
| wmo_pres (Pa) | $N_{\mathrm{TC}} \times 1$ | Minimum central pressure from the responsible WMO agency. |
| wmo_wind (m s$^{-1}$) | $N_{\mathrm{TC}} \times 1$ | Maximum sustained wind speed from the responsible WMO agency. |
| RO_datetime (seconds since 1 January 1970 00:00:0.0) | $N_{\mathrm{TC}} \times N_{\mathrm{maxRO}}$ | Date and time of RO profile. |
| RO_ID | $N_{\mathrm{TC}} \times 64 \times N_{\mathrm{maxRO}}$ | ID of collocated RO profile. |
| latRO (°N) | $N_{\mathrm{TC}} \times N_{\mathrm{maxRO}}$ | Latitude of mean RO tangent point. |
| lonRO (°E) | $N_{\mathrm{TC}} \times N_{\mathrm{maxRO}}$ | Longitude of mean RO tangent point. |
| QC | $N_{\mathrm{TC}} \times N_{\mathrm{maxRO}}$ | RO overall retrieval quality control (0 and 1 stand for good and bad profiles). |
| datediff_RO_TC (s) | $N_{\mathrm{TC}} \times N_{\mathrm{maxRO}}$ | Time difference between collocated RO profile and TC track position. |
| dist_RO_TC (km) | $N_{\mathrm{TC}} \times N_{\mathrm{maxRO}}$ | Distance between positions of TC track and mean RO tangent point. |
| bending_angle (rad) | $N_{\mathrm{alt}} \times N_{\mathrm{TC}} \times N_{\mathrm{maxRO}}$ | Ionosphere-corrected RO bending angle profile. |
| bending_angle_climatology (rad) | $N_{\mathrm{alt}} \times N_{\mathrm{TC}} \times N_{\mathrm{maxRO}}$ | Corresponding monthly climatological RO bending angle profile, interpolated from grid with 2.5° × 2.5° spatial resolution. |
| pressure (Pa) | $N_{\mathrm{alt}} \times N_{\mathrm{TC}} \times N_{\mathrm{maxRO}}$ | RO air pressure profile. |
| refractivity (N units) | $N_{\mathrm{alt}} \times N_{\mathrm{TC}} \times N_{\mathrm{maxRO}}$ | RO refractivity profile. |
| specific_humidity (kg kg$^{-1}$) | $N_{\mathrm{alt}} \times N_{\mathrm{TC}} \times N_{\mathrm{maxRO}}$ | RO specific humidity profile. |
| specific_humidity_climatology (kg kg$^{-1}$) | $N_{\mathrm{alt}} \times N_{\mathrm{TC}} \times N_{\mathrm{maxRO}}$ | Corresponding monthly climatological RO specific humidity profile, interpolated from grid with 2.5° × 2.5° spatial resolution. |
| temperature (K) | $N_{\mathrm{alt}} \times N_{\mathrm{TC}} \times N_{\mathrm{maxRO}}$ | RO air temperature profile. |
| temperature_climatology (K) | $N_{\mathrm{alt}} \times N_{\mathrm{TC}} \times N_{\mathrm{maxRO}}$ | Corresponding monthly climatological RO air temperature profile, interpolated from grid with 2.5° × 2.5° spatial resolution. |

https://doi.org/10.5194/essd-12-1-2020

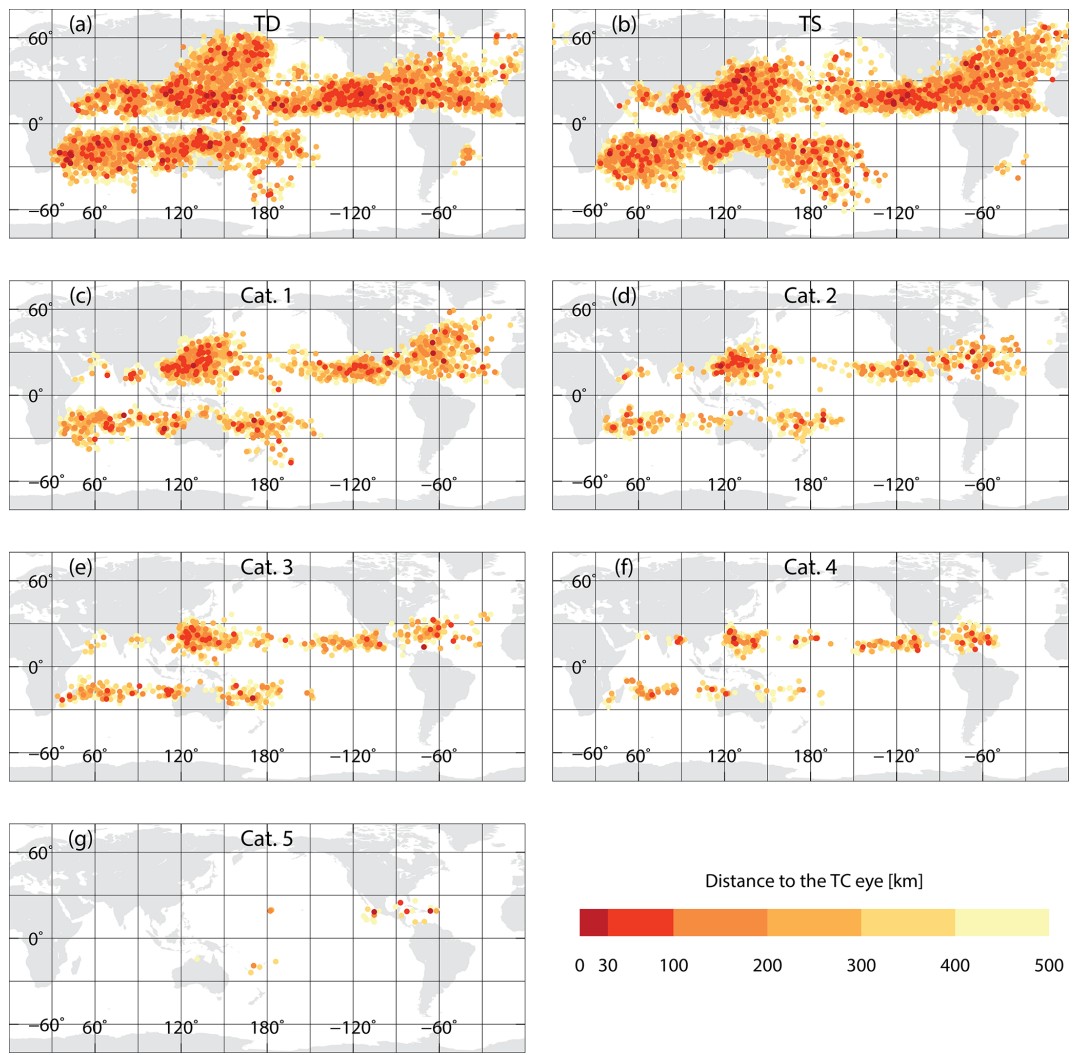

**Figure 2.** Map with distribution of RO profiles collocated with (**a**) tropical depressions, (**b**) tropical storms, (**c**) Category 1 TCs, (**d**) Category 2 TCs, (**e**) Category 3 TCs, (**f**) Category 4 TCs and (**g**) Category 5 TCs. Colours denote the distances between the RO profile and the nearest TC eye.

the yellow profiles indicate the TD final stages. Figure 4c represents the temporal and vertical behaviour for specific humidity. The RO data clearly reveal the temporal development of the storm's vertical structure. For the case of Hondo, we find that during the TC stages the mid-tropospheric thermal structure is warmer (warm TC inner core) and the upper troposphere is colder. Figure 5 shows how to use the full dataset, including information of the TC best tracks, the anomaly profiles and climatology profiles from RO for analysing the vertical thermal structure and understanding the behaviour of the storm. First, we use the TC best tracks to distinguish between the different storm stages (TD, TS and TC in Fig. 5). Then, we evaluate the anomaly profiles of the different RO variables bending angle, temperature and specific humidity, which have been computed by subtracting the reference monthly climatology profile in the respective

area from the individual profile. The anomaly profiles represent signatures created by the presence of the storm. Figure 5a shows the averaged bending angle anomalies for the TD, TS and TC status of Hondo. In the lower troposphere, a large negative anomaly in bending angle (relative to the climatology) is present due to the increase in humidity (Fig. 5c), while in the mid-troposphere the negative bending anomaly is due to the storm's warm core (Fig. 5b). In the upper troposphere, a positive anomaly in bending angle is caused by the cold cloud top. The TD moves less humidity than TS and TC (Fig. 5c). The warm core and cold cloud top are more distinct for TS and TC than for TD (Fig. 5b).

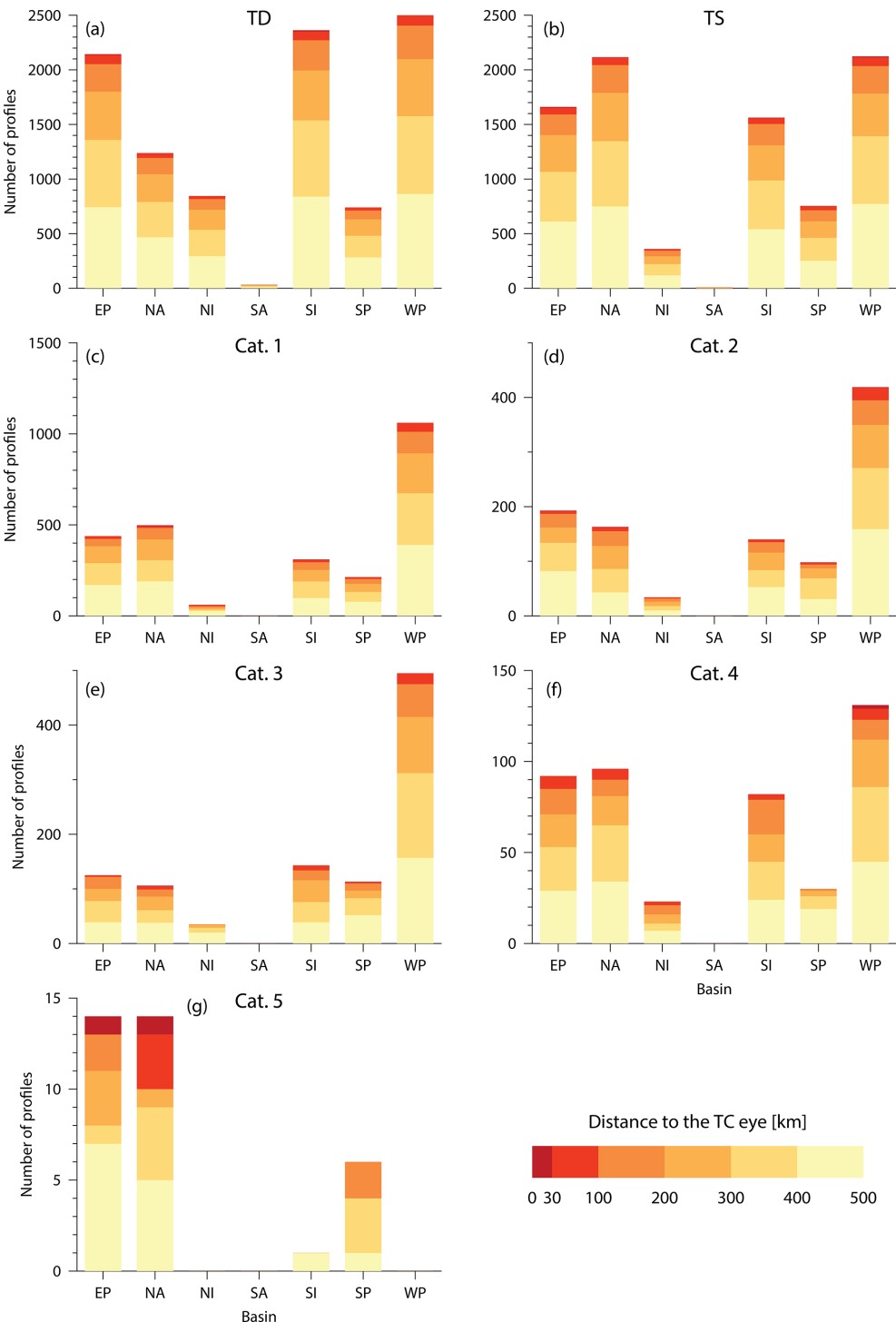

**Figure 3.** Histograms of collocated RO profiles with **(a)** tropical depressions, **(b)** tropical storms, **(c)** Category 1 TCs, **(d)** Category 2 TCs, **(e)** Category 3 TCs, **(f)** Category 4 TCs and **(g)** Category 5 TCs for different ocean basins. Colours denote the distances between the RO profile and the TC eye.

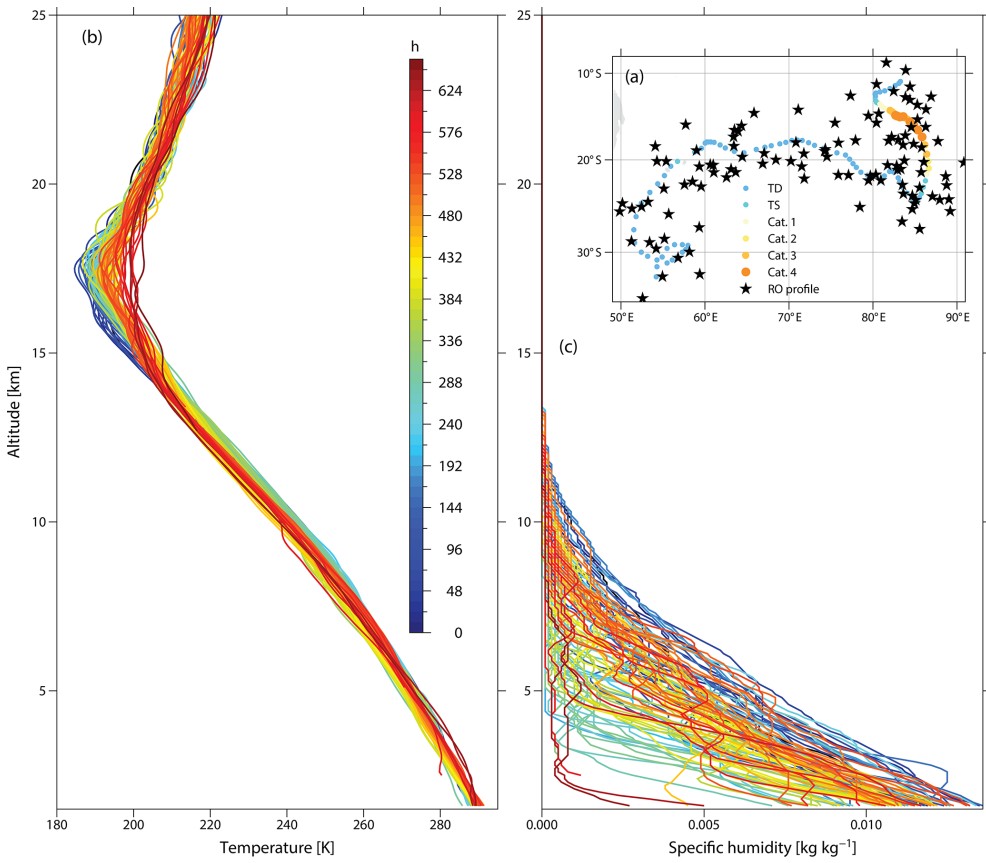

**Figure 4.** Temporal evolution of the Typhoon Hondo in 2008. Hondo best track and co-located ROs (**a**). Temperature (**b**) and specific humidity (**c**) profiles from the surface to 25 km of altitude since the beginning to the end of the storm.

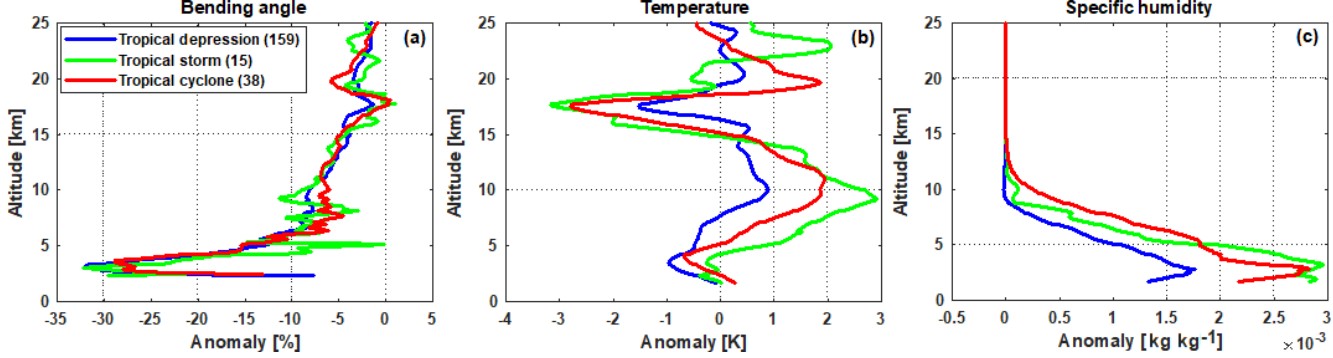

**Figure 5.** Averaged bending angle anomaly (**a**), temperature anomaly (**b**) and specific humidity anomaly (**c**) profiles for the TD, TS and TC status of Hondo 2008.

## 4  Data availability

All the data used to create this archive are publicly available. The WEGC GNSS RO record OPS v5.6 with high-quality atmospheric profiles is available online (https://doi.org/10.25364/WEGC/OPS5.6:2019.1, EOPAC Team, 2019) and archived at the Earth Observation Data Centre (EODC) in Vienna, where it can be downloaded via file transfer protocol (ftp) or secure file transfer protocol (sftp) according to the instructions and links also provided via the cited EOPAC Team (2019) DOI link CE1. Detailed information on the retrieval and on data quality is given by Angerer et al. (2017). The RO reference climatology was computed from OPS v5.6 profiles, which were averaged to a 2.5° × 2.5° latitude and longitude grid (each grid point containing profiles within a 300 km radius).

**Table 4.** Number of collocated RO profiles with TCs with regard to the acquisition year.

| Year | Number of profiles | Number of TCs | Number of TCs with at least 1 co-located RO profile |
|---|---|---|---|
| 2001 | 50 | 107 | 23 |
| 2002 | 222 | 100 | 56 |
| 2003 | 310 | 107 | 74 |
| 2004 | 335 | 105 | 73 |
| 2005 | 310 | 115 | 71 |
| 2006 | 1969 | 100 | 98 |
| 2007 | 4187 | 94 | 94 |
| 2008 | 5482 | 99 | 99 |
| 2009 | 5119 | 100 | 99 |
| 2010 | 3640 | 88 | 88 |
| 2011 | 3739 | 95 | 95 |
| 2012 | 3585 | 93 | 93 |
| 2013 | 4471 | 103 | 103 |
| 2014 | 3932 | 91 | 91 |
| 2015 | 4339 | 112 | 111 |
| 2016 | 2909 | 95 | 95 |
| 2017 | 2017 | 107 | 101 |
| 2018 | 1697 | 111 | 106 |
| Total | 48 313 | 1822 | 1570 |

**Table 5.** Number of collocated RO profiles with TC for different RO satellites.

| Satellite | Number of profiles |
|---|---|
| CHAMP | 1881 |
| CNOFS | 519 |
| F3C-FM1 | 6644 |
| F3C-FM2 | 3954 |
| F3C-FM3 | 1693 |
| F3C-FM4 | 4969 |
| F3C-FM5 | 5499 |
| F3C-FM6 | 4918 |
| GRACE-A | 1836 |
| GRACE-B | 562 |
| MetOp-A | 9416 |
| MetOp-B | 4704 |
| SAC-C | 1718 |

The monthly mean climatology was computed for the period for August 2006 to September 2017. The TC best tracks were obtained from the NOAA IBTrACS web page (https://www.ncdc.noaa.gov/ibtracs/, last access: 31 August 2020; Knapp et al., 2018) in CSV, netCDF or shapefile format for all available storms.

The created RO–TC dataset is available online at https://doi.org/10.25364/WEGC/TC-RO1.0:2020.1 (Lasota et al., 2020). It can be downloaded via file transfer protocol (ftp) or secure file transfer protocol (sftp), and the instructions for data download with different operating systems are also provided via the Lasota et al. (2020) DOI link cited at the beginning of this paragraph. The dataset consists of yearly folders, which refer to the year of the start of the storm. Each TC is saved in a separate file in NetCDF-4 format in the corresponding yearly directory. Filenames are self-explanatory with the format string NAME_year_IBTrACSuniqueID.nc. For example, file MERANTI_2016_2016253N13144.nc includes all atmospheric RO profiles co-located with the TC Meranti of IBTrACS ID 2016253N13144, which occurred in 2016. The description of the variables included in each dataset file can be found in Table 3 and the Supplement.

## 5 Discussion and conclusions

In this work, we provide a comprehensive archive of TCs' vertical structure for the period 2001–2018. Three main products are provided, co-located in time and space: global TC best tracks, RO profiles and RO climatological profiles. The archive can be used for different purposes for analysing the vertical thermodynamic structure of cyclones and the pre-cyclone environment.

The distance between the GNSS RO and TC best track is computed using as reference the RO mean tangent point coordinates, which usually corresponds to an altitude of about 12.5 km above the mean sea level, with a vertical resolution of about 100 m (Zeng et al., 2019) and a horizontal resolution of about 60 to 300 km (Gorbunov et al., 2004; Kursinski et al., 1997). The uncertainty given by the RO location must be summed to the TC best track position uncertainty, which mainly depends on the intensity of the storm and on the number and type of instruments used for monitoring (Landsea and Franklin, 2013). The TC best tracks are post-storm analyses relying on many different observations (ground-based, aircraft, satellite, radiosondes). The more intense the storm the more accurate the determination of the position is. The more observations are available the lower the uncertainty is. As an example, Landsea and Franklin (2013) report a position uncertainty of about 55 km (30 miles) for storms in the Atlantic Ocean observed just by satellite and a position uncertainty of about 15 km (8 miles) for major hurricanes observed by satellite, aircraft and ground-based instruments. In the worst case, the total co-location error between GNSS RO and TC best track could be up to 200 km in remote areas of the globe where just the satellite measurements are available and where storm intensities are low.

Part of this archive has already been used for studying the TC cloud top altitude (Biondi et al., 2013), to provide a characterization of the TC thermal structure and to detect TC overshooting for different ocean basins (Biondi et al., 2015). This demonstrates that, despite the uncertainties reported above, this archive is well suited for deepening our knowl-

**Table 6.** Number of collocated RO profiles with TCs with regard to the TC intensity and the distance to the TC eye on different ocean basins.

| Basin | Distance (km) | Intensity | | | | | | | | |
|---|---|---|---|---|---|---|---|---|---|---|
| | | Total | TD | TS | Cat. 1 | Cat. 2 | Cat. 3 | Cat. 4 | Cat. 5 | Not available |
| NA | 0–30 | 31 | 2 | 3 | 5 | 1 | 1 | 0 | 1 | 18 |
| | 31–100 | 280 | 43 | 69 | 10 | 7 | 6 | 6 | 3 | 136 |
| | 101–200 | 1021 | 149 | 254 | 63 | 27 | 13 | 9 | 0 | 506 |
| | 201–300 | 1698 | 253 | 443 | 114 | 42 | 25 | 16 | 1 | 804 |
| | 301–400 | 2243 | 324 | 597 | 116 | 43 | 23 | 31 | 4 | 1105 |
| | 401–500 | 2926 | 467 | 750 | 190 | 43 | 38 | 34 | 5 | 1399 |
| | Total | 8199 | 1238 | 2116 | 498 | 163 | 106 | 96 | 14 | 3968 |
| EP | 0–30 | 31 | 6 | 7 | 0 | 1 | 0 | 0 | 1 | 16 |
| | 31–100 | 346 | 85 | 62 | 15 | 5 | 3 | 7 | 0 | 169 |
| | 101–200 | 1109 | 252 | 188 | 40 | 25 | 22 | 14 | 2 | 566 |
| | 201–300 | 1920 | 442 | 338 | 93 | 28 | 22 | 18 | 3 | 976 |
| | 301–400 | 2666 | 616 | 455 | 120 | 52 | 39 | 24 | 1 | 1359 |
| | 401–500 | 3407 | 743 | 611 | 170 | 82 | 39 | 29 | 7 | 1726 |
| | Total | 9479 | 2144 | 1661 | 438 | 193 | 125 | 92 | 14 | 4812 |
| WP | 0–30 | 52 | 8 | 10 | 1 | 1 | 0 | 2 | 0 | 30 |
| | 31–100 | 567 | 98 | 79 | 48 | 23 | 20 | 6 | 0 | 293 |
| | 101–200 | 1722 | 307 | 252 | 119 | 45 | 60 | 11 | 0 | 928 |
| | 201–300 | 2787 | 521 | 390 | 219 | 79 | 103 | 26 | 0 | 1449 |
| | 301–400 | 4043 | 714 | 620 | 284 | 112 | 155 | 41 | 0 | 2117 |
| | 401–500 | 5139 | 863 | 773 | 390 | 159 | 157 | 45 | 0 | 2752 |
| | Total | 14310 | 2511 | 2124 | 1061 | 419 | 495 | 131 | 0 | 7569 |
| NI | 0–30 | 7 | 1 | 1 | 0 | 0 | 0 | 0 | 0 | 5 |
| | 31–100 | 88 | 27 | 15 | 8 | 2 | 0 | 2 | 0 | 34 |
| | 101–200 | 282 | 98 | 51 | 9 | 6 | 1 | 5 | 0 | 112 |
| | 201–300 | 486 | 184 | 72 | 10 | 8 | 5 | 5 | 0 | 202 |
| | 301–400 | 590 | 241 | 103 | 7 | 8 | 9 | 4 | 0 | 218 |
| | 401–500 | 823 | 294 | 118 | 27 | 10 | 20 | 7 | 0 | 347 |
| | Total | 2276 | 845 | 360 | 61 | 34 | 35 | 23 | 0 | 918 |
| SA | 0–30 | 0 | 0 | 0 | 0 | 0 | 0 | 0 | 0 | 0 |
| | 31–100 | 1 | 0 | 0 | 0 | 0 | 0 | 0 | 0 | 1 |
| | 101–200 | 16 | 7 | 1 | 0 | 0 | 0 | 0 | 0 | 8 |
| | 201–300 | 17 | 6 | 4 | 0 | 0 | 0 | 0 | 0 | 7 |
| | 301–400 | 27 | 9 | 3 | 0 | 0 | 0 | 0 | 0 | 15 |
| | 401–500 | 27 | 11 | 3 | 0 | 0 | 0 | 0 | 0 | 13 |
| | Total | 88 | 33 | 11 | 0 | 0 | 0 | 0 | 0 | 44 |
| SP | 0–30 | 13 | 2 | 2 | 1 | 1 | 1 | 0 | 0 | 6 |
| | 31–100 | 175 | 26 | 38 | 9 | 3 | 2 | 0 | 0 | 97 |
| | 101–200 | 538 | 81 | 101 | 27 | 7 | 13 | 1 | 2 | 306 |
| | 201–300 | 860 | 150 | 152 | 44 | 18 | 14 | 3 | 0 | 479 |
| | 301–400 | 1212 | 198 | 209 | 54 | 38 | 31 | 7 | 3 | 672 |
| | 401–500 | 1634 | 283 | 252 | 78 | 31 | 52 | 19 | 1 | 918 |
| | Total | 4432 | 740 | 754 | 213 | 98 | 113 | 30 | 6 | 2478 |
| SI | 0-30 | 38 | 12 | 5 | 2 | 0 | 0 | 0 | 0 | 19 |
| | 31–100 | 336 | 79 | 53 | 13 | 5 | 9 | 3 | 0 | 174 |
| | 101–200 | 1171 | 277 | 196 | 42 | 19 | 18 | 19 | 0 | 600 |
| | 201–300 | 1929 | 458 | 321 | 64 | 32 | 40 | 15 | 0 | 999 |
| | 301–400 | 2759 | 697 | 447 | 91 | 31 | 37 | 21 | 0 | 1435 |
| | 401–500 | 3296 | 840 | 541 | 98 | 53 | 39 | 24 | 1 | 1700 |
| | Total | 9529 | 2363 | 1563 | 310 | 140 | 143 | 82 | 1 | 4927 |
| Total | 0–30 | 172 | 31 | 28 | 9 | 4 | 2 | 2 | 2 | 94 |
| | 31–100 | 1793 | 358 | 316 | 103 | 45 | 40 | 24 | 3 | 904 |
| | 101–200 | 5859 | 1171 | 1043 | 300 | 129 | 127 | 59 | 4 | 3026 |
| | 201–300 | 9697 | 2014 | 1720 | 544 | 207 | 209 | 83 | 4 | 4916 |
| | 301–400 | 13 540 | 2799 | 2434 | 672 | 284 | 294 | 128 | 8 | 6921 |
| | 401–500 | 17 252 | 3501 | 3048 | 953 | 378 | 345 | 158 | 14 | 8855 |
| | Total | 48313 | 9874 | 8589 | 2581 | 1047 | 1017 | 454 | 35 | 24 716 |

edge of TCs. This is the first comprehensive archive collecting information of TC vertical structure, including profiles with a high vertical resolution from the surface to the TC cloud top and above, and providing high accuracy for all the main atmospheric parameters determining the development and the dynamics of the TCs.

This dataset allows gaining a better understanding of the TC inner structure especially in remote areas where ground-based sensors or radiosondes are not available and which are difficult to reach by aircraft. The independency of the ROs from the weather conditions provides a unique opportunity to profile extreme weather events without any risk and with global coverage.

The GNSS RO technique is well established, and the RO acquisitions are increasing thanks to the successfully launched COSMIC 2 mission, which will contribute to a better understanding of TCs, provide the necessary information to forecast the TC tracks with high accuracy and enable studying the diurnal changes of temperature during the extreme events. We believe that this archive is useful to get a better understanding of the TC development and intensification, as well as to increase our knowledge of the impact of TCs on the atmospheric structure.

**Supplement.** The supplement related to this article is available online at: https://doi.org/10.5194/essd-12-1-2020-supplement.

**Author contributions.** EL downloaded the data, developed the software and analysed the dataset. EL and RB designed the work and wrote the manuscript. RB supervised the project and acquired the funding. GK and AKS provided the RO data, supported the data archiving, contributed to the manuscript text and reviewed the manuscript.

**Competing interests.** The authors declare that they have no conflict of interest.

**Acknowledgements.** The work is accomplished in the frame of the VESUVIO project funded by the Supporting Talent in ReSearch (STARS) grant at Università degli Studi di Padova, IT. We thank Florian Ladstädter (WEGC) for providing the RO reference climatologies. We thank Armin Leuprecht (WEGC) for his support and guidance on all technical aspects of the archive files.

**Financial support.** This research has been supported by the Università degli Studi di Padova (VESUVIO).

**Review statement.** This paper was edited by David Carlson and reviewed by two anonymous referees.

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

## Remarks from the language copy-editor

CE1     I rephrased this sentence slightly to insert your changes. Please confirm.