# Peer review of "Tropical cyclones vertical structure from GNSS radio occultation: an archive covering the period 2001–2018"

_Earth System Science Data, 2020_

## Referee Comment (RC1) · Anonymous Referee #1 · 13 Jul 2020

Review of the study entitled "Tropical cyclones vertical structure from GNSS radio occultation: an archive covering the period 2001–2018" by ElÅijbieta Lasota et al.

First of all, I would like to express my deep respect to all the co-authors for creating this interesting data set and making it freely available for the public. This is quite interesting data set related to the GNSS RO and the tropical cyclones (TCs).

In this work, the authors provided a comprehensive data set regarding the TCs vertical structure from the co-located GNSS RO profiles for the period 2001-2018. The GNSS RO profiles are collected with the TC track information in time and space along with the background climatology. The presented data set is very useful for study and understanding the inner core structure of the TC. Personally, I have gone through the data link and downloaded the data. I also processed the data for a single cyclone from the data archive. The data link which is given in the manuscript is working properly and the data are freely accessible. I strongly believe that this data set can play a significant role in TC research and will become a benchmark data set for the future research. Overall the paper is well written, the created data set is demonstrated well and worth to be publishable. However, the manuscript needs some minor changes before the manuscript is ready for publication. Therefore, I strongly recommended the paper for publication in ESSD journal with some minor revision.

Major Comments/Suggestions: I have one major comment to the data set. When open the data link, I have found the data set in a specific folder with the name of individual years (2001, 2002,…2018). Some of the data files are having the name like 'NOT_NAMED_2001_2001031S13072.nc' in each folder of individual years. This will create some confusion for the users. It would be useful to provide sub folders with respect to each oceanic basin for each year. I strongly suggest to the authors, please include the subfolders with respect to the different ocean basin and keep the data files with respect to basins. I hope this may not take much time for the authors.

Specific Comments: Page 1 LN 24-25: It would be good to include one sentence related to the cyclone names over different basins. The authors used 'cyclones/storm/hurricanes' several times in the manuscript. It is good to introduce the cyclone names over different basins.

Page 1 LN 36: replace 'Numerical Weather Models (NWP)…..' to 'Numerical Weather Prediction (NWP)'

Page 1 LN 36-37: It is good to mention 'name of the oceanic basin'.

Page 3 LN 98: 'global monthly mean multi-satellite climatologies……' Is it the authors considered the data from 2001 to 2018? Please mention in the manuscript.

Page 3 LN 103: 'downloaded from The International. . ..' Change 'The' into 'the'.

Page 3 LN 115: change '6hour' to 6-hour.

Page 3 LN 125-130: 'seven levels based on the wind speed'. . . It would be good to include a table regarding the different types of TC intensity along with wind speed.

Page 4 LN 151-152: 'the minimum central pressure and the maximum sustained wind' Authors can mention here about 'wmo_pres' and 'wmo_wind' which are given in the data archive actually. . .. . .

Page 4 LN 167: 'We have collected 48313 co-locations between ROs and TCs from 1570 TCs'. The authors mentioned 1822 TCs in the abstract. Please check it once.

Page 5 LN 182: 'developed in the Indian ocean. . ...' Authors can specify the oceanic basin either it is North or South Indian Ocean. . .

Page 5 LN 196: 'reference climatology profile. . ..'Is it related to the climatology of the respective TC month?

Page 6 LN 225: 'corresponds to an altitude of about 15 km above the mean sea level' check it once.

Page 6 LN 246: The authors can include the usefulness of COSMIC-2 RO data, particularly the ability to study the diurnal changes of the temperature during the extreme events such as TCs/volcanic eruptions.

Figures: Figure 4: Use different color scale. Values more than 624 becomes white. It is difficult to identify.

Please check figure 5. Temperature and humidity anomalies up to 250 km? Is it really possible? I don't think so. Please correct the scale.

References: Ravindra Babu, S. and Liou, Y.-A.: Measurement report: Immediate impact of the Taal volcanic eruption on atmospheric temperature observed from COSMIC-

2 RO measurements, Atmos. Chem. Phys. Discuss., https://doi.org/10.5194/acp-2020-513, in review, 2020.

---

## Referee Comment (RC2) · Anonymous Referee #2 · 29 Jul 2020

Recommendation – acceptance subject to revision and clarification.

Ratings: Significance of the data set

Uniqueness - Low rating (1) As discussed in section 4 of the paper, the atmospheric RO profiles are available online, seemingly at two online locations. The Tropical cyclone best tracks data are also available online at the NOAA IbTrACS website. Hence many researchers may wish to extract the two data sets and match them themselves.

Usefulness. A reserved high rating (4). I believe there are many research issues and studies of tropical cyclone thermal and moisture structure that could be addressed with these data set. The reservation as to the usefulness is set out in comments below on

[Figure]

presentation quality

Completeness: High rating (4)

Data Quality. High rating (4). There is an extensive literature on the two primary data sets – the RO profiles and the ibTrACS cyclone tracks.

Presentation quality.

In my opinions, the utility of the data set for studying thermal and moisture structure of tropical cyclone is not well explained.

Lines 34 to 40 give references on the assimilation of the refractivity profiles (not the temperature and moisture profiles) in numerical weather prediction of tropical cyclones. Lines 40 to 69 give examples of research studies on tropical cyclones using RO profiles. However, these studies are almost exclusively related to detection of cloud height, to tropopause structure and to gravity wave generation in the upper atmosphere by tropical cyclones. The one paper referenced as examining of the vertical structure of cyclones is Biondi et all 2011a. That paper also has upper troposphere and lower stratosphere in the title, and its main findings arer on the bending angle of the radio occultation signal between 14 and 18 km. The paper does show temperature and moisture profiles for two cyclones through the depth of the troposphere. The paper makes the comment that the " water vapour anomalies from COSMIC agree largely with those of ECMWF, which can be explained by the fact that the ECMWF model is used in the derivation of the water vapour profiles"

Thus, so far from the literature review, there is no evidence concerning the quality of the thermal profiles and moisture profiles through the major structure of a tropical cyclone, which is in the troposphere. Line 70 of the paper states: "A comprehensive review on the use of RO observations to study TCs is given by Bonafoni et al. (2019)" The discussion in that paper is an expanded version of the discussion in lines 34 to 69 of the current paper.

[Figure]

There are two figures in the Review paper by Anthes et al 2011 showing RO profiles of moisture and temperature overlaid on dropsonde profiles, in typhoon Toraji and typhoon Jangmi. Both profiles are very impressive. However, whether the same methodology as that used in the current data set for obtaining moisture and temperature profiles from the RO refractivity profiles is unknown to this referee.

According to line 206, Detailed information on the retrieval and on data quality is given by Angerer et al. (2017). Referring to that paper, we learn that the calculation of physical variables (wet-temperature and specific humidity) requires a priori knowledge of the state of the atmosphere, for which ECMWF short-range forecasts are used. This immediately raises the issue as to how independent the resultant RO humidity profiles are from the ECMWF forecast profiles, the same issue that arises from the results in the Biondi et al 2011a paper. As opposed to that the paper by Kursinski et al JGR 1997 implies in section 2.3.3 that the derivation of the lower tropospheric water vapour profile requires only a background estimate of temperature from an independent source.

If I sound confused, I am. The main point is that all of this should be clarified for the tropical cyclones scientists who presumably are the potential users of the data set being documented. What is the quality of the RO temperature and moisture profiles in a tropical cyclone environment between about 100 hPa and the surface? How independent are the profiles from the ECMWF short-term forecast nearest profiles used in their derivation?

Given the wide-spread use of RO profiles in the last decade and the high profile of the program, I expect the issue is not with the data, rather it is with the level of explanation in the current write-up. Hence my recommendation of acceptance, subject to this issue being adequately resolved.

---

## Author Response (AR1)

**Reviewer 1**

>>> We would like to thank the anonymous reviewer for the insightful and constructive comments, which helped us to improve our manuscript. We appreciate the valuable comments and try to address the issues raised as best as possible.

According to the suggestions, we changed the data set structure. For each yearly folder, we created a set of subfolders with respect to the different ocean basin. However, the archive will be updated when the discussion will be closed so the paper remains consistent with the dataset structure.

Each major comment has been carefully considered point by point and responded below.

**Major Comments/Suggestions**

**I have one major comment to the data set. When open the data link, I have found the data set in a specific folder with the name of individual years (2001, 2002,...2018). Some of the data files are having the name like'NOT_NAMED_2001_2001031S13072.nc' in each folder of individual years. This will create some confusion for the users. It would be useful to provide sub folders with respect to each oceanic basin for each year. I strongly suggest to the authors, please include the subfolders with respect to the different ocean basin and keep the data files with respect to basins. I hope this may not take much time for the authors.**

>>> Thank you for the suggestion. We changed the data structure and every yearly folder is divided on subfolders related to the ocean basins. Hopefully, it makes the access to the dataset clearer for the users now. However, since a single TC can pass through many ocean basins, we assign every TC to the origin ocean basin, where the corresponding TC has started.

**Specific Comments**

**Page 1 LN 24-25: It would be good to include one sentence related to the cyclone names over different basins. The authors used 'cyclones/storm/hurricanes' several times in the manuscript. It is good to introduce the cyclone names over different basins.**

>>> We corrected the first sentence and explained the different names of tropical cyclones with respect to the basin of the origin. The first sentence of the introduction is now: "The Tropical Cyclones (TCs), known also as hurricanes in the North Atlantic Ocean and Northeast Pacific, typhoons in the Northwest Pacific and simply as cyclones in the South Pacific and Indian Ocean, are extreme weather events affecting the social lives of many people and the economy of entire countries."

**"Page 1 LN 36: replace 'Numerical Weather Models (NWP).....' to 'Numerical Weather Prediction (NWP)'**

>>> Corrected.

**Page 1 LN 36-37: It is good to mention 'name of the oceanic basin'.**

>>> We added the information, the sentence now reads as "Huang et al. (2005), for the first time, assimilated RO refractivity profiles to forecast the Typhoons Nari in 2001 and Nakri in 2002, which which developed in the North Western Pacific Ocean.".

**Page 3 LN 98: 'global monthly mean multi-satellite climatologies......' Is it the authors considered the data from 2001 to 2018? Please mention in the manuscript.**

>>> Information added. The sentence now reads as "Furthermore, we made use of global monthly mean multi-satellite climatologies processed by the WEGC (based on OPSv5.6 profiles for data betweenin the period 2001- and 2017)."

**Page 3 LN 103: 'downloaded from The International....' Change 'The' into 'the'.**

>>> Corrected.

**Page 3 LN 115: change '6hour' to 6-hour.**

>>> Corrected.

**Page 3 LN 125-130: 'seven levels based on the wind speed'...It would be good to include a table regarding the different types of TC intensity along with wind speed.**

>>> Thank you for the suggestion. We replaced the information in the text with the table qualifying TC intensity based on the wind speed (Figure 1).

**Page 4 LN 151-152: 'the minimum central pressure and the maximum sustained wind' Authors can mention here about 'wmo_pres' and 'wmo_wind' which are given in the data archive actually......**

>>> We added information rephrasing as "The TC is described by the basin of development, the name of responsible recording WMO agency, the distance of the TC from land, the date, time and coordinates at each 6-hour best track stage, the nature of the storm, the storm translation speed, the minimum central pressure and the maximum sustained wind speed provided by the responsible WMO agency and stored in 'wmo_pres' and 'wmo_wind' variables, respectively."

**Page 4 LN 167: 'We have collected 48313 co-locations between ROs and TCs from1570 TCs'. The authors mentioned 1822 TCs in the abstract. Please check it once.**

>>> Thank you for the remark. The numbers are different because in the abstract we refer to the number of TCs collected from IBTrACS (1822), while 1570 is the number of TCs for which we found at least 1 co-located RO. For 252 TCs we did not find any co-location. This is now explained in the manuscript.

**Page 5 LN 182: 'developed in the Indian ocean.....' Authors can specify the oceanic basin either it is North or South Indian Ocean...**

>>> The sentence is now: "… which developed in the South Indian ocean …"

**Page 5 LN 196: 'reference climatology profile....'Is it related to the climatology of the respective TC month?**

>>> Yes, it is the monthly climatology profile. The sentence is now: "… by subtracting the reference monthly climatology profile in the respective area from the individual profile."

**Page 6 LN 225: 'corresponds to an altitude of about 15 km above the mean sea level' check it once.**

>>> We double checked it. We were wrong, we should have written 'of about 12.5 km above the mean sea level'. We corrected this sentence.

**Page 6 LN 246: The authors can include the usefulness of COSMIC-2 RO data, particularly the ability to study the diurnal changes of the temperature during the extreme events such as TCs/volcanic eruptions.**

>>> Thank you for the suggestion, we added the relevant information: "The GNSS RO technique is well established and the RO acquisitions are increasing thanks to the successfully launched COSMIC 2 mission, which will contribute to a better understanding of TCs, provide the necessary information to forecast the TC tracks with high accuracy and enable studying the diurnal changes of temperature during the extreme events."

**Figures: Figure 4: Use different color scale. Values more than 624 becomes white. It is difficult to identify.**

>>> We have changed the colour scale.

**Please check figure 5. Temperature and humidity anomalies up to 250 km? Is it really possible? I don't think so. Please correct the scale.**

>>> Our apologizes, it was a mistake, the values should have been divided by 10. Now, it is corrected (Figure 2).

**References: Ravindra Babu, S. and Liou, Y.-A.: Measurement report: Immediate impact of the Taal volcanic eruption on atmospheric temperature observed from COSMIC-2 RO measurements, Atmos. Chem. Phys. Discuss., https://doi.org/10.5194/acp-2020-513, in review, 2020.**

>>> This paper refers to volcanic clouds, we prefer to focus on TC. We think that the sentence reported before "The GNSS RO … enable studying the diurnal changes of temperature during the extreme events." Is enough to explain the importance of COSMIC-2.

**Reviewer 2**

>>> We thank the reviewer for the careful review of the manuscript. We appreciate the valuable comments and try to address the issues raised as best as possible.

According to the suggestions, we improved the Introduction section adding the information about studying thermal and moisture TC structure using radio occultation profiles. We hope, we clarified the reviewer's concerns on retrieval of RO meteorological profiles.

**Recommendation – acceptance subject to revision and clarification.**

**Ratings: Significance of the data set**

**Uniqueness - Low rating (1) As discussed in section 4 of the paper, the atmospheric RO profiles are available online, seemingly at two online locations. The Tropical cyclone best tracks data are also available online at the NOAA IbTrACS website. Hence many researchers may wish to extract the two data sets and match them themselves.**

**Usefulness. A reserved high rating (4). I believe there are many research issues and studies of tropical cyclone thermal and moisture structure that could be addressed with these data set. The reservation as to the usefulness is set out in comments below on presentation quality .**

**Completeness: High rating (4)**

**Data Quality. High rating (4). There is an extensive literature on the two primary datasets – the RO profiles and the ibTrACS cyclone tracks.**

**Presentation quality.**

**In my opinions, the utility of the data set for studying thermal and moisture structure of tropical cyclone is not well explained.**

**Lines 34 to 40 give references on the assimilation of the refractivity profiles (not the temperature and moisture profiles) in numerical weather prediction of tropical cyclones. Lines 40 to 69 give examples of research studies on tropical cyclones using RO profiles. However, these studies are almost exclusively related to detection of cloud height, to tropopause structure and to gravity wave generation in the upper atmosphere by tropical cyclones. The one paper referenced as examining of the vertical structure of cyclones is Biondi et all 2011a. That paper also has upper troposphere and lower stratosphere in the title, and its main findings are on the bending angle of the radio occultation signal between 14 and 18 km. The paper does show temperature and moisture profiles for two cyclones through the depth of the troposphere. The paper makes the comment that the " water vapour anomalies from COSMIC agree largely with those of ECMWF, which can be explained by the fact that the ECMWF model is used in the derivation of the water vapour profiles "**

**Thus, so far from the literature review, there is no evidence concerning the quality of the thermal profiles and moisture profiles through the major structure of a tropical cyclone, which is in the troposphere. Line 70 of the paper states: "A comprehensive review on the use of RO observations to study TCs is given by Bonafoni et al. (2019)" The discussion in that paper is an expanded version of the discussion in lines 34 to 69 of the current paper.**

**There are two figures in the Review paper by Anthes et al 2011 showing RO profiles of moisture and temperature overlaid on dropsonde profiles, in typhoon Toraji and typhoon Jangmi. Both profiles are very impressive. However, whether the same methodology as that used in the current data set for obtaining moisture and temperature profiles from the RO refractivity profiles is unknown to this referee.**

>>> We thank the reviewer for pointing this out. We extended the Introduction section adding more relevant studies on TC thermodynamic structures using radio occultation temperature and humidity

profiles. The paper Biondi et al. 2011a was one of the first studying the thermodynamic structure of the TC and, as highlighted by the reviewer, focusing on the UTLS. At these altitudes the moisture information is almost completely coming from the model, however at lower altitudes the information from the model is balanced by the information coming from the RO signal.

Anthes et al. (2011) used profiles processed by COSMIC Data Analysis and Archival Center with 1D-Variational algorithm and background information from NCEP-NCAR reanalysis. The retrieval scheme in the lower troposphere was statistically based, where the respective contributions of the background temperature and humidity change with latitude and altitude. In this study we use the Wegener Center OPSv5.6 with background information from ECMWF short term forecast. The details of this tropospheric retrieval scheme have recently been described by Li et al. (2019), which we now cite as well. These authors have not only introduced the OPSv5.6 moist air retrieval algorithm in detail but also intercompared it with the UCAR/COSMIC and EUMETSAT ROM SAF retrievals, overall showing that in the lower to middle troposphere the moisture information is predominantly coming from the RO data, and carefully implemented state-of-the-art retrievals all lead to highly comparable temperature and humidity retrieval results. The recent work of Rieckh et al. (2018), which we as well cite now, corroborates these findings from intercomparing tropospheric humidity profiles from four retrievals (incl. OPSv5.6) and with radiosonde and further profiles. This information has been added in section 2.1 "RO profiles".

We extended the introduction with the study conducted by Anthes et al. (2003), Vergados et al. (2013), and Chen et al (2020). The manuscript now contains the new sentences:

"In a recent study, Chen et al. (2020) assimilated RO data during the genesis of 10 TCs in the North Western Pacific Ocean in the period 2008-2010. The results confirmed the benefit of assimilation of RO refractivity improving the humidity estimation in the lower and the middle troposphere and thus the forecast of the TCs."

"Anthes et al. (2003) were the first, comparing RO soundings with radiosonde observations during the typhoon Toraji 2001. The results revealed that the RO temperature profiles were consistent within 1 K, whilst RO water vapour observations tended to be slightly drier than radiosonde measurements above the middle troposphere. A similar agreement between RO and dropsonde observations was presented by Anthes et al. (2011) for the typhoon Jangmi 2008."

"Vergados et al. (2013) for the first time used more of 1500 RO temperature, water vapour and refractivity profiles to study the moist thermodynamic structure in the lower and the upper troposphere of 42 North Atlantic TCs in the period 2002-2010. The analysis showed that the RO observations are able to capture the dimension, eyewall and rainbands of the TCs at different stages. Especially, the gradual decrease and wavelike pattern of water vapour was observed with increasing distance from the TC centre. Furthermore, a drop of WV was noticed in the lower and upper troposphere when the TCs develop from a tropical depression to Category 1 intensity."

**According to line 206, Detailed information on the retrieval and on data quality is given by Angerer et al. (2017). Referring to that paper, we learn that the calculation of physical variables (wet-temperature and specific humidity) requires a priori knowledge of the state of the atmosphere, for which ECMWF short-range forecasts are used. This immediately raises the issue as to how independent the resultant RO humidity profiles are from the ECMWF forecast profiles, the same issue that arises from the results in the Biondi et al 2011a paper. As opposed to that the paper by Kursinski et al JGR 1997 implies in section 2.3.3 that the derivation of the lower tropospheric water vapour profile requires only a background estimate of temperature from an independent source.**

**If I sound confused, I am. The main point is that all of this should be clarified for the tropical cyclones scientists who presumably are the potential users of the data set being documented. What is the quality of the RO temperature and moisture profiles in a tropical cyclone environment between about 100 hPa and the surface? How independent are the profiles from the ECMWF short-term forecast nearest profiles used in their derivation?**

**Given the wide-spread use of RO profiles in the last decade and the high profile of the program, I expect the issue is not with the data, rather it is with the level of explanation in the current write-up. Hence my recommendation of acceptance, subject to this issue being adequately resolved**

>>> Refractivity profiles can be straightforwardly transformed to dry temperature and dry pressure profiles using reduced refractivity equation in the regions where water vapour is negligible (usually above about 10 km of altitude) and ideal gas and hydrostatic equilibrium assumptions can be applied. We now cite as well the paper by Scherllin-Pirscher et al. (2011) that explains this well in a basic concise form in its Section 5 therein; Li et al. (2019) that is also cited now (see above) then deepens this information on moist air retrieval vs. dry air retrieval with rich detail.
Briefly, in the lower troposphere, the dry-air assumption is no longer valid due to the presence of abundant water vapour. Hence, ancillary information about temperature, pressure or water vapour pressure is required to calculate the physical atmospheric parameters. The commonly applied solutions encompass an iterative method using GNSS RO refractivity and independent temperature profile, or one-dimensional variational (1-DVar) retrieval method, where the background information from a weather model is merged to the RO profile. Li et al. (2019) introduce all these approaches in more detail and includes a detailed description of the Wegener Center algorithm as noted in the answer above. The Wegener Center OPSv5.6 uses the ECMWF short-term forecast as background information. The respective contributions of the background temperature and humidity change with latitude and altitude. The retrieval to a priori error ratio (RAER) separately for temperature and humidity, describes the amount of the background information contained in the statistically optimized profile at each altitude level and this information is originally stored in the WEGC OPSv5.6 products. Li

et al. (2019) illustrate some examples of how background and RO observations go together in the troposphere.

We update the section 2.1 "RO profiles" with a brief description of the methodology and the relevant references:

[revised manuscript text omitted]